# Toward a Better Understanding of Metal Nanoparticles, a Novel Strategy from *Eucalyptus* Plants

**DOI:** 10.3390/plants10050929

**Published:** 2021-05-07

**Authors:** Hanadi Sawalha, Rambod Abiri, Ruzana Sanusi, Noor Azmi Shaharuddin, Aida Atiqah Mohd Noor, Nor Aini Ab Shukor, Hazandy Abdul-Hamid, Siti Aqlima Ahmad

**Affiliations:** 1Laboratory of Bioresource Management, Institute of Tropical Forestry and Forest Products (INTROP), University Putra Malaysia, UPM, Serdang 43400, Selangor, Malaysia; hanadisawalha4@gmail.com (H.S.); ruzanasanusi@upm.edu.my (R.S.); aidaatiqah.an@gmail.com (A.A.M.N.); dnoraini@upm.edu.my (N.A.A.S.); hazandy@upm.edu.my (H.A.-H.); 2Department of Biochemistry, Faculty of Biotechnology and Biomolecular Sciences, University Putra Malaysia, UPM, Serdang 43400, Selangor, Malaysia; noorazmi@upm.edu.my; 3Department of Forestry Science and Biodiversity, Faculty of Forestry and Environment, University Putra Malaysia, UPM, Serdang 43400, Selangor, Malaysia; rambod.abiri@upm.edu.my; 4Institute of Plantation Studies, University Putra Malaysia, UPM Serdang 43400, Selangor, Malaysia

**Keywords:** *Eucalyptus*, nanoparticles, metal nanoparticles, biosynthesis, nanotechnology

## Abstract

Nanotechnology is a promising tool that has opened the doors of improvement to the quality of human’s lives through its potential in numerous technological aspects. Green chemistry of nanoscale materials (1–100 nm) is as an effective and sustainable strategy to manufacture homogeneous nanoparticles (NPs) with unique properties, thus making the synthesis of green NPs, especially metal nanoparticles (MNPs), the scientist’s core theme. Researchers have tested different organisms to manufacture MNPs and the results of experiments confirmed that plants tend to be the ideal candidate amongst all entities and are suitable to synthesize a wide variety of MNPs. Natural and cultivated *Eucalyptus* forests are among woody plants used for landscape beautification and as forest products. The present review has been written to reflect the efficacious role of *Eucalyptus* in the synthesis of MNPs. To better understand this, the route of extracting MNPs from plants, in general, and *Eucalyptus*, in particular, are discussed. Furthermore, the crucial factors influencing the process of MNP synthesis from *Eucalyptus* as well as their characterization and recent applications are highlighted. Information gathered in this review is useful to build a basis for new prospective research ideas on how to exploit this woody species in the production of MNPs. Nevertheless, there is a necessity to feed the scientific field with further investigations on wider applications of *Eucalyptus*-derived MNPs.

## 1. Introduction

Nanotechnology is a multidisciplinary, modern, and innovative generation of applied science [1] concerned with the application, synthesis, and design of devices and materials on the nanoscale (smaller than 100 nm) [2]. The earliest discussion of the possible consequences of nanotechnology can be traced back to 1986, when Eric Drexler developed some nanotechnology techniques along with other technological advances [3]. Over the past years, the advances of nanotechnology have been significantly sparked in the fields of medical, agriculture, and forestry [4,5,6]. To develop an efficient synthesis of nanoparticles (NPs), different biological, chemical, and physical strategies have been applied [7,8]. The synthesis of NPs using conventional methods may increase the environmental pollution and toxicity risk due to the production of by-products [9,10]. Recently, more interest has been gained, focusing on environment-friendly strategies to produce NPs [11,12]. Unlike the mentioned chemical and physical techniques, the biological strategy relies on employing living organisms such as plants, fungi, and bacteria to synthesize the metal nanoparticles (MNPs) [7,13]. The synthesis of engineered MNPs is mainly divided into two modes as seen in Figure 1. The first approach is a top-down synthesis technique, where the large bulk materials are segmented into NPs [14]. In contrast, a bottom-up (self-assembly) technique entails an arrangement of small fragments into more complex structures [15]. Chemical and biological methods of synthesis commonly use the bottom-up approach in which the obtained MNPs are chemically homogeneous and relatively pure [16].

Interestingly, green synthesis and the characterization of green MNPs have been applied using plant resources supplemented with different metals including zinc (Zn), silver (Ag), selenium (Se) [8], titanium (Ti), platinum (Pt) [17], nickel (Ni), gold (Au), and copper (Cu) [5]. The plant-based MNPs offer different properties such as antioxidant [18], anti-inflammatory, antidiabetic [19], anticancer, antimicrobial, and immunomodulatory ones [20]. Several previous papers reported the synthesis of MNPs from different plant extracts including coffee and tea [21], sorghum [22], peanut [23] and *Hordeum vulgare* [24], due to their efficacy as alternatives for chemicals.

This review discusses the synthesis of MNPs from *Eucalyptus* (family Myrtaceae), an evergreen woody plant native to many regions in the world [25]. *Eucalyptus* is a good exporter of bioactive molecules such as terpenoids and phenols, and its organs have been perfectly used in the green route of MNPs synthesis [26]. Thus, this comprehensive review summarizes the most important literature about the synthesis of MNPs using *Eucalyptus* species. Moreover, the factors that influence the production process, characterization techniques, and potential recent applications are also highlighted.

## 2. Green Synthesis of Metallic Nanoparticles: A Reliable Plant Route—Technique for a Cleaner Future

Over the decades, advances in MNPs synthesis, characterization, and application have been dramatically increased [27]. The most popular methods for the synthesis of MNPs are chemical and physical methods; nevertheless, these traditional methods are not affordable on a large scale and release toxic compounds, which determined their uses [28,29]. To counteract this problem, a novel, safe, and sustainable experimental method, known as the green method or biosynthesis of NPs, was developed [30]. This method interconnects nanoscience and the field of biotechnology [28]. The application of green MNPs has allowed the replacement of harmful substances with agents that are nonhazardous to the ecosystem [31]. This mechanism provides a reliable [32], simple, inexpensive, and relatively renewable technique with reproducibly produced materials [30]. Furthermore, this kind of synthesis employs the living cells of multiple biological resources such as plants [33], algae [34], fungi, and bacteria [35], as can be observed in Figure 2, and the synthesized NPs can be stored at room temperature [36]. These entities are nanofactories that provide an eco-friendly method to produce MNPs with a broad range of morphological and physicochemical features [37].

As mentioned above, the application of plants as a green synthesis bioreactor is an exciting frontier implemented in MNP technology [38]. Various plant extracts have been considered as alternative and eco-friendly agents for replacing the use of chemical compounds [39] owing to their reasonable cost and the unpretentious maintenance needed [35]. Meanwhile, plant extracts can detoxify heavy metals in a short time as the accumulation of small traces are toxic and cause environmental pollution [40]. What distinguishes the use of plant extracts from other biological agents is that other organisms go through complex procedures to maintain microbial cultures [15]. However, procuring plant extracts using the metal ion solution under certain conditions is a simple explanation of the prompt steps to obtain the green MNPs [41]. Moreover, the kinetics of the biological synthesis route is higher than that of other biosynthetic ways parallel to chemical nanoparticle elaboration [35].

Different plant parts could be used in the biosynthesis of MNPs such as root, shoot, leaf, and fruit due to their synthesis of super phytochemicals [42]. The active chemical agents in these parts include polyphenols, polyols [43], phenolic acids, terpenoids, sugars, alkaloids, and proteins [44]. Chemically, these secondary metabolites are complex but are environment-friendly and play a significant role in reducing and stabilizing metallic ions [37]. Micronutrients existing in plants and their derivatives are called phenols [45]. Phenols in the plant extracts are the main compounds for the synthesis and stabilization of NPs [46]. Salgado et al. [45] reported that phenolic compounds act as effective precursors during the formation of FeNPs from *Eucalyptus*. Polyphenols are favorable as anti-oxidants, anticancer agents, and antimicrobials [47]. The oxygenated terpenes formed are terpenoids, which are among the biggest natural product families present in all organisms [48]. Essential oils are the source from which plant terpenoids are mainly acquired [49]. The aforementioned metabolites are factors acting as metal salt reductants in addition to capping materials in the process of synthesizing MNPs from plants [50] as presented in Figure 3.

According to Vitta et al. [51], the green approach to MNP production is summarized as an oxidation–reduction reaction that takes place due to the reductive capacity of the components inside or outside the cell. In turn, these components transfer electrons in the form of MNPs with a neutral charge (zero). The functional groups contributing to the conversion of metal ions into nanoparticles are ketones, carboxyl, alcohols, amines, sulfhydryl, and aldehydes, and the presence of any of these groups in any biological compound will demonstrate the success of the conversion process [52]. In addition, many studies have been conducted to produce MNPs using plant materials such as *Eucalyptus globulus* [26], *Peganum harmala* [53], *Lawsonia inermis* [54], *Azadirachta indica* [55], *Crocus sativus* [56] and *Piper longum* [57].

## 3. *Eucalyptus* in a General View

*Eucalyptus* forests are the largest cultivated and natural hardwood flora worldwide, a genus belonging to the Myrtaceae family [58] (Figure 4). *Eucalyptus* genus includes more than 700 species distributed around the world [59]. For example, Brazil [58], Australia [60], Egypt [61], China [62], Portugal, the European Union, and Spain [63] are the countries where *Eucalyptus* plantations are widely distributed in their landscapes. As a general description, the *Eucalyptus* genus comprises perennial and evergreen plants that vary from small shrubs to tall trees [64].

*Eucalyptus* leaves exhibit a dimorphism in shape in the early and mature stages. Juvenile leaves are almost oval, sessile, and sometimes glaucous, while the leaves in the mature stage are completed, oblong, oval, and firm [65]. Different kinds of *Eucalyptus* bark are observed, e.g., permanent, deciduous, coarse, and soft. Bark traits such as fiber longitude, stiffness, color, and furrows differ according to the age of the plant [66]. *Eucalyptus* blossoms show different forms, i.e., axillary umbel, panicle, corymb, or cyme. Petals together appear like a cup shape (operculum), which are occasionally variable (such as acute, obtuse, or horned) within the species of the genus, and their stamens are colorful when mature [67]. *Eucalyptus* fruits are woody and have different shapes including globular, conical, or hemispherical [68]; while, their seeds have a size from 1 mm to 2 cm and are spherical or maybe elliptical [69].

*Eucalyptus* has tremendous benefits, as this genus has great economic importance due to its fast-growing nature [62], tolerance to biotic stress conditions [5], and versatile uses of its essential oils [70]. *Eucalyptus* essential oils have aromatic scents and can be extracted from various species of *Eucalyptus* [71]. Ecologically, *Eucalyptus* oils act as herbivore deterrents, alleviate ozone toxicity, and mitigate the temperature throughout the time of fires [60]. An excess of over 70% of 1,8-cineole in *Eucalyptus* makes it suitable as a pest repellent. For example, Indian *E. globulus* essential oils are targeted for medicinal purposes due to their activity against bacteria, fungi, and oxidative agents [72]. Additionally, *Eucalyptus* oils are useable in food additives, fragrances, and pharmaceutical products [73].

Seeds of *E. globulus* are exploited as a source of porous carbon, which proves its performance as a highly efficient supercapacitor and energy storage material [74]. *Eucalyptus* wood is a naturally renewable resource consisting of regularly ordered cells that contain different quantities of lignin, cellulose, and hemicelluloses [75]. It is commonly used as a fundamental material of cellulose pulp, timber, and panels [76]. Furthermore, for its corrosion resistance, *Eucalyptus* wood is largely used as a source of fuel in the construction and paper industries [62]. Moreover, wood sawdust from *Eucalyptus* trees is the pure substance from which activated carbon powder is made [77].

Bioactive compounds with antimicrobial activity involving flavonoid, tannin, phenolic, phloroglucinol, terpenoid, and cardiac glycosides could be found in abundance in *Eucalyptus* species [61]. Although the bark of *Eucalyptus* is detached and considered as a waste, it is a good fuel source [76]. Plenty of tannins are found in the bark of *E. urophylla*, *E. grandis,* and *E. citriodora*, which are necessary to form wood adhesive materials [76]. Due to its excellent sorption efficiency, *Eucalyptus* bark is effective to handle effluents contaminated with chromium (Cr) [78]. Gao et al. [70] notified that *Eucalyptus* bark and its derivatives can be used to clean polluted hydrous solutions.

The main biomolecules existing in *E. globulus* leaf extracts are tannins and flavonoids [79], which gives the possibility of use in pharmaceutical and agricultural areas [63]. A vast range of biomolecules is contained in the leaves of *E. camaldulensis* comprising tannins, proteins, flavonoids, saponins, and carbohydrates, which are important in the process of NP formation [80]. Synthesis of NPs from different parts of *Eucalyptus* trees has been extensively outlined in the literature. The presence of polyphenols such as flavonoids in *E. robusta* leaf extracts is responsible for iron NP (FeNP) fabrication [51]. Nickel oxide NPs (NiONPs) synthesized from *E. globulus* leaves are beneficial in protecting human health due to their antibacterial activity [81]. While zinc oxide NPs (ZnONPs) have been greenly synthesized as antioxidants and photocatalysts from *E. globulus* leaves [25]. These characteristics demonstrate that *Eucalyptus* is a valuable genus that could be used for many purposes.

## 4. Synthesis of MNPs from *Eucalyptus*

The history of the first MNPs derived from *Eucalyptus* goes back to 2008 when Ramezani et al. [82] reported the use of *Eucalyptus* extract to produce gold NPs (AuNPs); the objective of their study was to develop a new structure for a nontoxic and environmentally friendly technique. A few years later, scientist’s attention emerged regarding this plant due to its efficiency in producing MNPs. Thus far, several *Eucalyptus* species have been adopted to produce MNPs such as *E. chapmaniana* [32], *E. globulus* [25,26,79,83,84,85,86], *E. leucoxylon* [87], *E. oleosa* [88,89], *E. camaldulensis* [90,91], *E. robusta* [51], and *E. tereticornis* [92].

In the last few years, ample information has been collected and reported related to the synthesis of MNPs from *Eucalyptus*. Several kinds of MNPs with different shapes, sizes, and characterizations have been synthesized from *Eucalyptus* such as silver NPs (AgNPs) [93], gold NPs (AuNPs) [94], FeNPs [95], and nickel NPs (NiNPs) [81]. As shown in Table 1, AgNPs are the most popular among MNPs synthesized from *Eucalyptus*, while few studies have been conducted on the synthesis of AuNPs, FeNPs, NiNPs, and bimetallic Fe/NiNPs. *Eucalyptus* leaves have been vastly used for the synthesis of MNPs [51,91,96,97] while the bark and wood are rarely used. The use of *Eucalyptus* leaves to produce MNPs is due to their richness in active substances such as tannic acids, volatile oils, flavonoids, and organic acids [93]. Furthermore, polyphenols in *Eucalyptus* leaves act as antioxidants with the ability to control NP aggregation and as capping agents that improve MNP dispersion [98]. The abundant amount of phenolic compounds and antioxidant proportion in *Eucalyptus* leaves would be a determinant factor in synthesizing diverse MNPs [98,99,100]. On the other hand, the bark of *Eucalyptus* is neglected and considered as waste. Despite this, it has abundant amounts of polyphenolic substances to produce MNPs [101]. Different characterization tools have been used to reveal chemical and morphological features. In terms of the sizes and shapes of the MNPs, sizes are varied according to the selected procedure, while their shapes look spherical and cubic in most experiments (Table 1).

### 4.1. Extraction and Synthesis of MNPs from Eucalyptus

From the literature, the methodology of generating and reaching the final output of MNPs comprises several steps (Figure 5). The first step is preparing the *Eucalyptus* extract, on which the synthesis of NPs depends. *Eucalyptus* parts are collected, followed by rinsing and drying several times, then the dried materials are powdered and added to deionized water. The resultant extract was then centrifuged for a few minutes and filtered to obtain the final volume [72]. According to Chand et al. [39], the second step of synthesis is carried out by adding a relevant metal salt solution to the extract, keeping it at a certain temperature for a specific time until the color of the solution changes, which is an evidence of the reduction of the metal ion and NP formation. The produced particles are then centrifuged, washed and dried for use and characterization.

### 4.2. Other Green Extraction Routes

Plants have thousands of bioactive compounds used in various sectors, therefore it is crucial to quickly evolve and develop high-yield extraction techniques [110]. However, conventional ways require a long extraction period, which may cause thermal phytodecomposition [111]. Moreover, through conventional methods, some volatiles could be lost and toxic solvent residues could be produced [112]. In recent years, new extraction techniques have been developed and are being used as alternatives to conventional methods with low efficiency including supercritical fluid extraction, subcritical water extraction, microwave-assisted extraction (hydro-distillation), and pressure solvent.

*Eucalyptus* trees are beautiful and usually cultivated for their gum, timber, oil, and pulp [113]. Among various products derived from this plant, essential oils and extracts are the most attractive renewable resources [114]. A variety of complex compounds such as sesquiterpenes, ketones, terpenoids, esters, alcohols, aromatic phenols, 1,8-cineole, ethers, monoterpenes, and oxides are found in *Eucalyptus* oil [115]. *Eucalyptus* oil has great biological properties (such as antimalarial, anti-allergenic, antiseptic, and anti-asthmatic ones), making it suitable to be used in pharmaceuticals. It has also been used in cleaning, cosmetic and food products [116], and nanotechnology [117].

#### 4.2.1. Supercritical Fluid Extraction

Supercritical fluid extraction (SFE) is a green chemical method that has been attracting considerable interest as a sustainable technique in daily food, fragrance, and pharmaceutical industries [118]. Carbon dioxide (CO_2_) is the popular solution used in SFE and has special physical and chemical properties including being nontoxic, non-flammable, and affordable. Moreover, it can be stored at low pressure and close to room temperature [114]. Using CO_2_ in SFE does not leave any residue; therefore, it offers an oil of high quality. In contrast with conventional extraction methods, SFE utilizing CO_2_ needs low temperatures to operate, which maintains the thermally labile ingredients in the extracts [119]. Moreover, CO_2_ is a non-polar compound; thus, it is not effective for extracting polar molecules. Introducing a co-solvent such as ethanol, water, or methanol increases the amount of the extraction yield [100].

SFE positively impacts the environment as it employs no, or only slightly, environmentally harmful organic solvents. The performance of SFE operation, as well as the oil yield, are impacted by various factors including extraction time, temperature, sample size and operating pressure [120]. Zhao et al. [114] reported that the extraction pressure and time are increased along with the increase in oil yield. Additionally, they demonstrated that the temperature moderately affected the extraction and reacted with pressure. Hence, it is important to adjust the right extraction parameters to eliminate any co-extraction between compounds [121]. SFE has been carried out to extract various compounds from different plant species. *E. globulus* bark [100,120], olive leaves [122] and *Cannabis sativa* [123] are some examples employing SFE as an extraction method.

#### 4.2.2. Subcritical Water Extraction

Extraction with Subcritical water (SW) is a mechanism mainly dependent on water as an extractant at relatively high pressure to maintain the liquid form and temperature ranging from 100 °C to 374 °C [112]. SW facilitates fast extraction operation and utilizes low temperatures, which prevent losses of volatiles as well as the degradation of compounds [124]. The SW technique includes a transmission of solutes from the sample to the extraction medium utilizing convection, diffusion and partitioning equilibrium [125]. Water offers unrivalled characteristics under subcritical conditions. Eco-friendly, affordable and rapid extraction rate are the main advantages of this process [126]. Many reports have investigated the use of water as a solvent for several components including lignocelluloses [127]. Moreover, it has been used in carbonization, liquefaction and gasification processes [128].

SW has been employed to extract several compounds from *Eucalyptus* species. According to Wu et al. [128], the ideal condition of SW extraction oil is under 300 °C for 30 min, which is enough to obtain 30.1% of bio-oil. In addition, Kulkarni et al. [126] reported the use of SW for the extraction of antioxidants from *E. grandis*. Jimenez-Carmona and de Castro [112] isolated *Eucalyptus* essential oil by using SW technique.

#### 4.2.3. Subsubsection Microwave-Assisted Extraction

Based on the previous studies, microwave-assisted (MAE) is an extraction method adopted to obtain organic materials, inorganic and pesticide remnants from natural products [129,130]. It is an unconventional process used as an alternative mechanism due to its shortened extraction time, improved quality yield [131], and consumption of a low quantity of solvents [132]. Farhat et al. [133] highlight that the use of microwaves for extracting bio-oils is inexpensive and harmless to the environment. The MAE technique focuses on the absorption of microwave irradiation by the plant matrix’s water [134]. The efficiency of the MAE method can be influenced by the material size, material-to-solvent ratio, extraction time, and power level [135]. Various bioactive compounds have been extracted from aromatic plants using microwave technology such as *Ocimum basilicum* [136], *Cymbopogon martini* [137], and *Cinnamomum camphora* [138]. Bhuyan et. al. [139] extracted phenolic compounds and flavonoids from *E. robusta* via MAE. Saoud et al. [140] found that exposure to microwaves for 3 min would be enough to extract *Eucalyptus* essential oil. Gharekhani et al. [141] obtained phenols and flavonoids from *E. camaldulensis,* and Soet [131] also used this technique to extract *E. citriodora* oil.

#### 4.2.4. Ultrasound-Assisted Extraction

Ultrasound-assisted extraction (UAE) is a widely used method that serves as an excellent physicochemical condition for the treatment of raw materials [142]. By means of ultrasound, the extraction procedure can be boosted by intensifying the transfer of heat mass between the solvent and plant matrix [143]. Such a technique is being advanced due to its ability to enhance the quality and environmental health of the extraction procedure [134]. UAE is an alternative method that has the potential to reinforce industries such as wood processing [144] and natural product extraction [145]. Furthermore, UAE shows many advantages including selectivity, low energy consumption, high yield quality, lessened hazardous compounds, and high automation level [146]. The application of UAE has been discussed by many studies. For instance, Rodrigues and Pinto [147] employed UAE to extract phenolic compounds from coconut. Goula [148] extracted seed oil of pomegranate. Alissandrakis et al. [149] extracted volatile compounds from the flowers and honey of citrus. Gullon et al. [63] proved that UAE is an ecofriendly and inexpensive method for the extraction of antioxidant phenols from *E. globulus* leaves. Meanwhile, Xu et al. [150] found that there is a great potential to use UAE in the process of hemicellulose separation from *E. grandis*.

### 4.3. Factors Influencing Biological Synthesis of MNPs from Eucalyptus

The efficiency of establishing the process of MNP synthesis is controlled by several parameters. Studies have reported the influence of pH [151,152,153], reactant concentration [152], reaction temperature [153], and reaction time [154] on MNP formation. The effects of the above-mentioned parameters on the formation of MNPs from *Eucalyptus* are explained in the following sections.

#### 4.3.1. Effect of pH

The acidity and basicity of the medium have a major role in the formation of NPs in different ways such as influencing the surface morphological structure, precipitation position, reduction reaction rate [153,155], and stability [154] and contributing to the change of plant metabolite charge [156]. Spherical and decahedral MNPs are well fabricated under a higher degree of pH [17]. Specification of the pH value is usually done by a pH meter [98]. It is worth noting that few studies have addressed pH despite its significant role in the process of MNP synthesis [156]. For example, the preparation of AgNPs from *E. globulus* [85] studied different values of pH (2.7, 7.0, 10.0), showing an effect on AgNP distribution size. A study by Liu et al. [95] also reported the formation of FeNPs from *Eucalyptus* using different pH values (4,6,8) and, thus, achieved the optimum removal efficiency of hexavalent chromium from hydrous solution at pH 4.

#### 4.3.2. Effect of Reactant Concentration

Metal ion concentrations are varied according to the parameters taken through the synthesis. Dubey et al. [154] found that a large particle size of AgNPs and AuNPs is attained by the higher concentration of the metal ion. Pourmortazavi et al. [88] found that increasing from 1 to 5 mmol/L of metal ion concentration showed a larger size of AgNPs, while enhancement of the concentration to 10 mmol/L achieved an opposite result. Pinto et al. [85] observed that a very low concentration of *Eucalyptus* bark extraction plays a significant role in AuNP size in which the lower concentrations increase the particle size. Briefly, the addition of 50 mg/L of gold to *E. globulus* leaf extract and essential oil was investigated in the preparation of AuNPs [94]. More-supplemented *Eucalyptus* extraction has led to the increase of the synthesis reaction rate in research conducted to prepare FeNPs from *E. globulus* [157].

#### 4.3.3. Effects of Reaction Time

Studies have also been focused on the reaction time since increasing and reducing time gives different results. Reaction time mainly depends on the concentration of the extract [85]. Further, it is responsible for the change of medium color, which is an indication of NP formation [158]. Biosynthesis of NPs is known to be speedier than conventional methods as the formation of NPs starts in a few minutes [85,154]. Nonetheless, the period required to show the changing of color differs among the experiments based on the operation factors [88].

A study has been conducted by Rahimi-Nasrabadi et al. [87] on the synthesis of AgNPs using *E. leucoxylon*. The researchers have been appraised that the higher the reaction time, the sharper the peak. Other studies confirmed that the color changed to brown in a few minutes [32,87,93,96,97]. Vitta et al. [51] indicated the formation of FeNPs after 30 min from the addition of the extraction until the appearance of the changed color (black). Pourmortazavi et al. [88] concluded that increasing synthesis hours would decrease the AgNPs’ size.

#### 4.3.4. Effects of Reaction Temperature

Temperature is one of the critical factors in the synthesis process that has a significant role in influencing the morphology of NPs [153]. NP size decreases with a higher reaction temperature [154,159]. During the biosynthesis of MNPs, prior to the change in color, metal ions are exposed to a certain temperature that varies according to the NPs and the procedure followed. AgNPs present a smaller size as a result of promoting the temperature of *Eucalyptus* extract [88]. Golmoraj et al. [106] examined the stability of AuNP created from *E. camaldulensis* using three temperatures (4, 25, and 45 °C) and demonstrated that stability was achieved at all three.

## 5. Characterization Techniques of Metal NPs Derived from *Eucalyptus*

Within the sequence of steps in NP biosynthesis, characterization is no less important than the rest of the process, as it is the way to determine the size and morphological structure of the particles [14]. Characterization of MNPs is carried out by various techniques [160] such as UV–visible spectroscopy (UV-vis) [91], scanning electron microscopy (SEM), Fourier transmission infrared spectroscopy (FTIR), transmission electron microscopy [31], X-ray diffraction (XRD), energy dispersive X-ray spectrometer (EDX), and dynamic light scattering (DLS) [39]. The most popular techniques used to characterize MNPs derived from *Eucalyptus* and the functions of each technique are summarized in Figure 6.

### 5.1. UV-Vis

The general use of UV-vis is to inspect the shape and size of NPs in aqueous solutions as well as to determine the optical properties of that solutions [32,161,162], reaction time [154], stability, and formation of NPs [103,105,160,163]. The UV-vis tool has particular advantages such as being simple, rapid, susceptible, and appropriate for several types of NPs [164,165]. In short, it is a process that measures the quantity of absorbed radiation via an ingredient in solution [160]. In UV-vis, the valence band and the conduction band, take their place where the movement of the electrons is free, then a band of surface plasmon resonance is produced according to the resonance oscillation of NPs with the wave of fallen light [166,167]. A wide range of studies has reported that each MNP has a special absorption band. According to the literature, AgNPs produce bands at about 200–800 nm that exist in the UV-vis spectrum, which is utilized to characterize NPs in a range of 2 to 100 nm [160]. This confirms the result by Sulaiman et al. [32], which recorded the surface plasmon resonance at 413 nm. A study done by Mo et al. [93] investigated the wavelength of synthesized AgNPs from *Eucalyptus* leaf ranging from 400 to 500 nm. Moreover, 300 to 400 nm was the wavelength recorded by UV-vis through the synthesis of NiNPs from *E. globulus*. Single wavelength peaks of *E. camaldulensis*–mediated AgNPs started from 449 nm and gradually increased by extending the reaction time [80].

### 5.2. SEM

SEM is an electron microscopy technique that gives the speed and high resolution of a direct surface image besides measuring the dimensions (size and shape) at micro- and nanoscales [168,169,170]. SEM employs a high beam of energetic electrons, which are focused onto the surface of NP specimens and go through a scattering process; observed signals from these scattered electrons provide the final characteristics of the materials [160]. It has been noted that SEM can investigate exterior structure; besides, it can provide worthy information regarding the particle aggregation degree and purity [171].

The commencement of SEM revealed that under optimum conditions, AgNPs derived from *Eucalyptus* have around 50 nm diameter [87]. Furthermore, SEM clearly observed the spheroidal shape of FeNPs from *Eucalyptus* leaf extracts [98]. SEM confirmed changes in cell surface structure and the deactivation of the important hospital-acquired pathogen cell membranes by using green synthesized AgNPs [104].

### 5.3. TEM

TEM is a powerful imaging technique used to estimate the morphological properties and size of NPs [103,170,172,173]. TEM interacts with the specimen via a beam of electrons to pattern the image on the photographic plane [174]. The remarkable use of TEM is to form a high-resolution image of the NPs’ atomic space [175]. In contrast with SEM, TEM provides higher image accuracy and extra analytical measurements [176]. An image of the cross-sectional area for synthesized AgNPs from *Eucalyptus* leaf extract has been obtained utilizing TEM [87]. TEM has been adopted to show the morphological structure of AgNPs [93] and FeNPs [95,107] mediated by *Eucalyptus*.

### 5.4. FTIR

FTIR is a robust, simple, and widespread analytical technique [177], applied to screen and evaluate whether or not the biological molecules of NPs are taking part in the synthesis [160,178]. Over and above, this method has been utilized to search for the interactivity between enzymes and their substrates at the time of the catalytic process [179]. FTIR uses infrared radiation (IR) in which some are passing through the sample and some are absorbed, followed by a formation of the spectrum. The function of this spectrum is to make the molecular fingerprint, which in turn determines the identity of the sample [180]. FTIR is used to characterize the formation of FeNPs synthesized from *Eucalyptus* and to detect the biomolecules responsible for causing the metal reduction and NP capping [98]. The leaf extract of a *Eucalyptus* sample has been examined using FTIR spectral analysis to reveal the biomolecules that acted as reducing and capping agents through the synthesis process [109]. Mo et al. [93] used FTIR to point out the potential biomolecules accountable for the reduction and stabilization during the synthesis of AgNPs. FTIR was served to depict the leaf extract before and after the synthesizing of FeNPs from *Eucalyptus* [95]. Pourmortazavi et al. [88] used FTIR to examine how the plant extract is influenced by the extraction temperature and to analyze the interaction of AgNPs synthesized using *Eucalyptus* extract. The results observed that, while increasing the temperature, the peak intensity decreased. Golmoraj et al. [106] studied the surface chemical composition of AuNPs mediated using *E. camaldulensis* methanol extract by FTIR. In this context, the functional carboxyl group was detected, which has a key role in the stability of AuNPs.

### 5.5. XRD

XRD is the most popular technique used to investigate the chemical compound as well as determine the crystal structure [32,107,181] and size of MNPs [182]. It is worth noting that, through XRD, shape features relating to different materials such as glasses, high conductors. and biomolecules could be explored [8]. Revealing of the MNPs’ characteristics by XRD goes through specific steps starting from casting a beam of X-rays on the crystals, then the fallen beam dispersed via the atoms, which leads to the emergence of diffraction patterns followed by the interference of beams with each other [160]. Bragg’s Law could be used to detect the interference to observe the crystal material’s characteristics [183]. Domain size of the crystal material can be calculated using the Scherrer formula [184]. Diffraction patterns are crucial since the analysis of crystal materials relies on them; these patterns imply the physical characteristics and chemical composition of the materials [185]. XRD confirmed that the nature of synthesized FeNPs mediated using *Eucalyptus* leaf extract is predominantly amorphous [98]. Rahimi-Nasrabadi et al. [87] examined the purity level and the composition of AgNPs synthesized from *Eucalyptus* via XRD. Mo et al. [93] reported the crystalline structure of AgNPs derived from *Eucalyptus* using XRD. Meanwhile, Ali et al. [81] demonstrated the crystalline structure of NiNPs by XRD recorded patterns. Additionally, Wu et al. [186] confirmed the crystalline structure of FeNPs derived from the hybrid species of *E. urophylla* and *E. grandis*.

## 6. Recent Applications of *Eucalyptus*-Mediated MNPs

Due to their novelty and marvelous characteristics, NPs have been applied in different essential aspects of research and life, encompassing chemical analysis, environmental remediations, medical tools, and agricultural sciences [187]. These characteristics came from their surface charge and area, as well as the size and shape [188]. MNPs have proven their effectiveness in the anticancer therapeutic system, textile manufacturing, food processing, and the treatment of contaminated water [188]. MNPs obtained from *Eucalyptus* species have been studied in several areas and continue to offer blooming advances.

### 6.1. Environmental and Agricultural Applications

Environmental remediation applications have been reported by many researchers. The capacity of biosynthesized FeNPs mediated using *Eucalyptus* leaf extract to remediate eutrophic wastewater has been confirmed [98]. In a recent study, Wu et al. [186] examined the efficiency of FeNPs mediated through *Eucalyptus* to eliminate arsenic from polluted areas, which added a piece of knowledge about the fabricated NPs as they have high adsorption capability and considerable environmental application regarding the removal of arsenic. Similarly, the high efficiency of calcined bimetallic Fe/PdNPs using *Eucalyptus* leaves to purify contaminated drink water resources from arsenic (III) has been observed [108]. In the latest novel study, AgNPs were well fabricated from the gum-based *E. camaldulensis*. The authors of the study demonstrated that AgNPs proved to be great antibacterial and antifungal candidates in cleaning the wastewater system and liquid residues from dyes [189].

In the field of renewable energy, AgNPs have been prepared from *E. globulus* in combination with *Azadirachta indica* and *Coriandrum sativum*. In addition, AgNPs are effective antireflectors when wrapped around the outer layer of a silicon solar cell [190]. The MNPs biosynthesized using *Eucalyptus* extracts are also exploited in the field of agriculture. *E. officinalis* is utilized in preparing AgNPs, where the produced NPs can increase rice productivity by exhibiting a nematocidal entity to dominate root and soil nematode growth [191]. It is also worth noting the development of using *Eucalyptus* nanoextracts as pesticide and fungicide products to increase the agricultural production in a safe and a sustainable way. For example, FeNPs are applicable to be exploited in pest management against *Rhizoctonia solani*, *Botrytis cinerea,* and *Fusarium oxysporum* besides the antifeedant activity against diamondback moth [192]. Furthermore, the high percentage of 1,8-cineole in *E. globulus* extract makes it very suitable to be used in pest management as a botanical insecticide against the aphid species *Myzus persicae* [193].

### 6.2. Medical and Pharmaceutical Applications

Nanoscience has an important role in modern medical technology, especially in cancer treatment. For instance, AgNPs prepared from *E. tereticornis* demonstrated an antibacterial effect to be used in the industry of antibacterial drugs, antioxidant efficiency, and anticancer therapeutics to resist MCF-7 cancer cell lines [96]. *E. camaldulensis*-based AgNPs are effective against certain cancer cell lines in addition to displaying antimicrobial activity [194]. AuNPs green synthesized utilizing *E. tereticornis* have recently been reported to enter the world of medical and pharmaceutical activity. This is due to their antioxidant, antibacterial, and anti–human breast cancer activities [195]. Syukri et al. [196] observed that the covering of surgical sutures with AgNPs mediated using *E. camaldulensis* is a successful procedure to be used for treating wounds infected by bacteria. Paosen et al. [104] concluded that AgNPs mediated using *E. citriodora* possess a great antimicrobial-resistance against serious hospital-gained diseases. Another investigation by Radwan et al. [101] using AgNPs from *E. camaldulensis* confirmed anti-aging properties, where they blocked apoptosis and hindered the formation of skin wrinkles. A novel medical discovery by Salih et al. [197] aiming at investigating the effects of AgNPs biosynthesized using *E. globulus* versus a worm parasite (*Echinococcus granulosus*) concluded that the synthesized NPs performed as a rapid sporicidal agent toward parasites, which are the leading cause of human liver cysts. The antioxidant influence of FeNPs prepared from *E. robusta* opens a window to new pharmacological applications [51].

## 7. Conclusions

MNPs green synthesized via *Eucalyptus* have been applied in various primary research and practical life aspects due to their antibacterial, antioxidant, and anticancer effects. The current review showed that the green synthesis of MNPs is a cleaner route after considering its nontoxicity, simplicity, sustainability, and eco-friendly advantages. In simple terms, the main focus of the present review was on the mechanism for extracting MNPs from *Eucalyptus,* which was based on standard steps starting from collecting plant parts to adding the metal salt solution and centrifugation to obtain MNPs. Furthermore, the roles of micronutrients existing in *Eucalyptus* have been displayed such as phenols and their derivatives in the synthesis method. Likewise, this review has summarized some factors (pH, reactant concentration, reaction time, and temperature) impacting the synthesis and the most popular characterization tools (TEM, SEM, UV-vis, XRD, and FTIR) used to reveal the characteristics of *Eucalyptus*-mediated MNPs. Although *Eucalyptus* is used for producing MNPs, it is a versatile genus that could be used for various purposes, and there is a need for further investigations in applications such as agriculture and forestry. To the best of the authors’ knowledge, it will expand scientist’s perceptions about the synthesis of MNPs from *Eucalyptus*, thus enriching future empirical findings. Nonetheless, the interesting information reported in the literature about the synthesis of MNPs from *Eucalyptus* indicates a promising potential to follow the pattern of its cleared and discovered properties for further applications in various aspects. Hence, it would be easy to build and develop a nonpathogenic biological environment that can manufacture MNPs.

## Figures and Tables

**Figure 1 plants-10-00929-f001:**
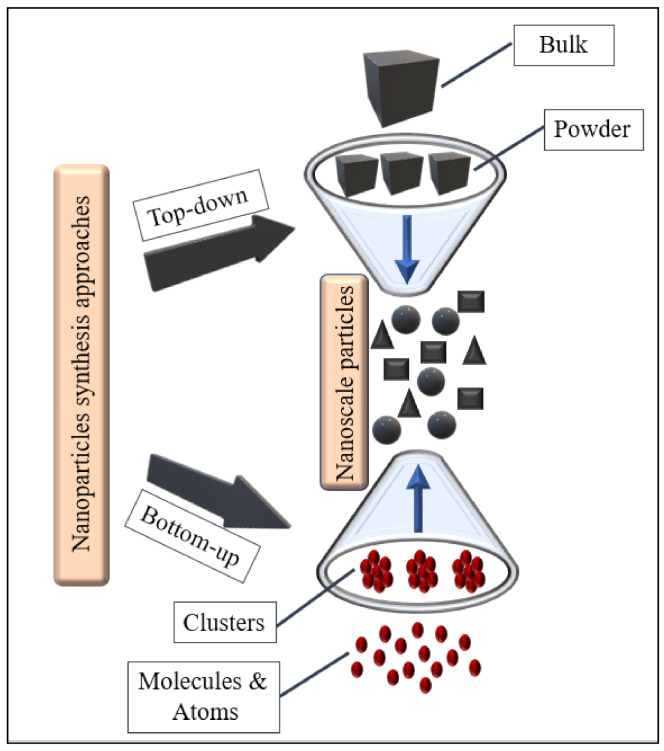
A simplified interpretation of the bottom-up and top-down ways. For the biosynthesis of metal nanoparticles (MNPs), the bottom-up method is used where small atoms and molecules are combined into clusters and then to nanoscale particles. Contrariwise, in the top-down method, the nanoparticles (NPs) are created from the bulk materials.

**Figure 2 plants-10-00929-f002:**
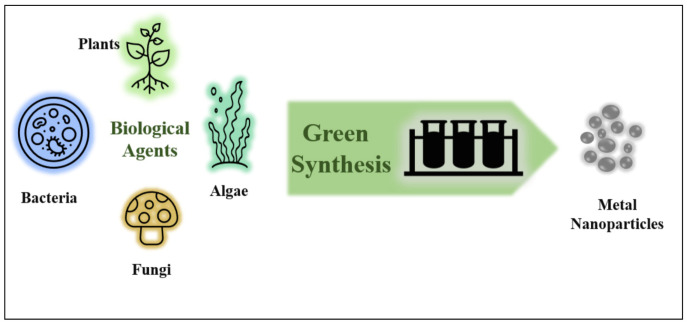
The green synthesis method using several unicellular (bacteria, algae) and multicellular (plants, fungi) biological agents—a system in which these agents are utilized to generate nanostructures of metals in one step.

**Figure 3 plants-10-00929-f003:**
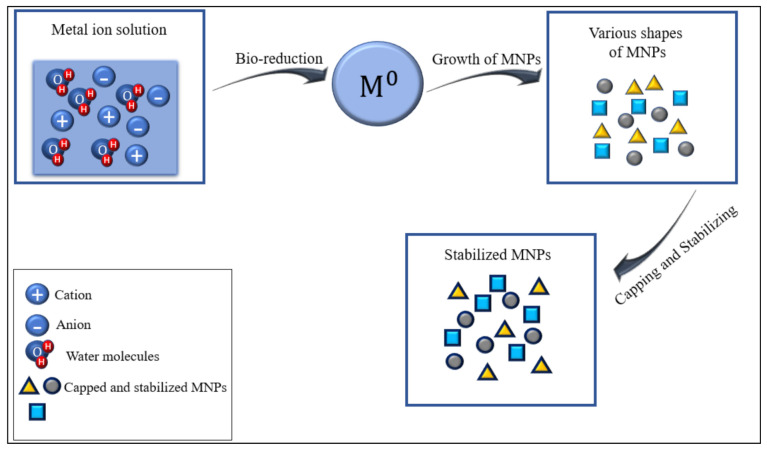
Chemical reduction and stabilization of MNPs. Reduction of the metal salt solution is done by the presence of secondary metabolites in plant extracts such as (phenols, polyphenols, and flavonoids) to maintain zero-valent metals. Then, MNPs with various shapes and sizes are yielded, followed by coating and stabilizing with capping agents (secondary metabolites).

**Figure 4 plants-10-00929-f004:**
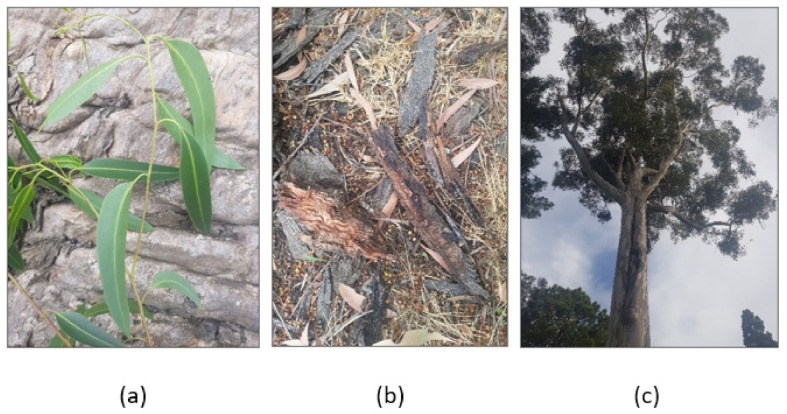
Photos of (**a**) *Eucalyptus cladocalyx* (leaf), (**b**) *Eucalyptus pilularis* (bark), (**c**) *Eucalyptus cladocalyx* (whole tree). Photos are taken from the “Royal Botanic Gardens Victoria—Melbourne Gardens”.

**Figure 5 plants-10-00929-f005:**
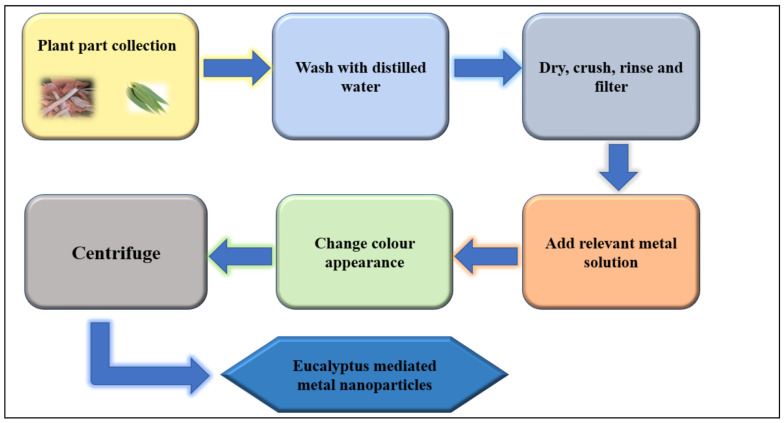
Schematic mechanism of synthesis MNPs from *Eucalyptus*. Starting from the collection of plant parts, then going through several steps, and ending with filtration to reach the extraction. After that, the extract is used to reach the targeted MNPs after mixing it with the metal ion solution to observe the changed color, which indicates the procedure’s success.

**Figure 6 plants-10-00929-f006:**
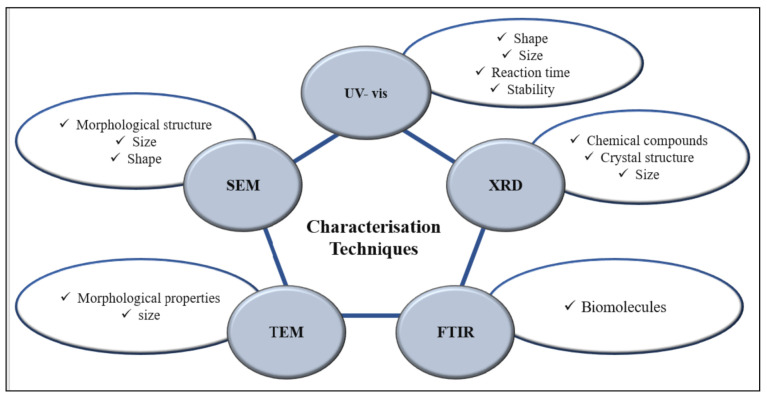
The most-used characterisation techniques (TEM, SEM, UV-vis, XRD, and FTIR) to investigate different features (size, shape, diameter) of MNPs derived from *Eucalyptus*.

**Table 1 plants-10-00929-t001:** *Eucalyptus*-mediated synthesis of metallic NPs.

NP	Plant Part	Characterization Tools	NP Size (nm)	NP Shape	References
Ag	leaf	UV, FTIR, SEM, TEM, XRD,	10–30	spherical	[91]
	leaf	UV, XRD, SEM, TEM, EDS	14.5	monocyclic, cubic	[96]
	bark	TEM	468.7	spherical	[101]
	leaf	UV, SEM, EDX	12.3–14.43	spherical	[97]
	bark	UV, XRD, FESEM, HRTEM, TEM, FTIR	25	spherical	[102]
	leaf	UV, FETEM, XRD, FTIR	4–60	spherical	[93]
	leaf	UV, XRD	60	cubic	[32]
	leaf	UV, XRD, SEM, TEM	50	cubic	[87]
	leaf	UV, SEM, EDX, FTIR	21	spherical	[88]
	leaf	UV, FTIR, TEM, SEM	8–15	spherical	[103]
	leaf	SEM	17.51	spherical	[104]
	leaf	UV, SEM, DLS, NTA, EDS	12	spherical	[80]
	wood	UV, XRD, TEM, FTIR, FESEM, EDAX	25–30	spherical	[105]
	leaf	UV, XRD, SEM, EDS, TEM	14.5	spherical, agglomerated	[96]
Au	bark	UV, TEM, DLS	~20	triangular, hexagonal, spherical	[85]
	leaf	UV, TEM, FTIR	1.25–17.5	spherical	[106]
	leaf	TEM	12.8 ± 6.3	spherical	[94]
Fe	leaf	UV, homemade device	0.2–2	spherical	[51]
	leaf	SEM, EDS, FTIR, XRD	20–80	spherical	[98]
	leaf	XRD, FTIR, XPS, GCMS	~95	spherical	[95]
	leaf	SEM, TEM, XRD	70 ± 10	spherical	[107]
Calcined-Fe/Pd	leaf	TEM, EDS, XRD, XPS	30–60	spherical	[108]
Bimetallic Fe/Ni	Leaf	FTIR, TG, SEM, EDS, XRD, XPS	20–50	spherical and irregular	[109]

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
