# Peer review of "Toward a Better Understanding of Metal Nanoparticles, a Novel Strategy from Eucalyptus Plants"

_plants, 2021, doi:10.3390/plants10050929_

Round 1
Reviewer 1 Report
General comments.
In the paper “Towards a Better Understanding of Metal Nanoparticles, a Novel Strategy from Eucalyptus Plant” authors reported a review based on researches concerning different organisms to manufacture MNPs. In particular, the review reports the efficacious role of Eucalyptus in the synthesis of MNPs.
Paper is interesting, and god write. However, some other references can be added. I suggest minor revision
Specific comments.
Page 9 line 296
Based on the previous studies, microwave-assisted (MAE) is an extraction method adopted to obtain organic materials and pesticide remnants from natural products [129].
Please, add in the sentence microwave-assisted (MAE) is an extraction method adopted to obtain organic materials, inorganic and pesticide remnants from natural products [129].
In this context please add reference
Wood pellets for home heating can be considered environmentally friendly fuels? Heavy metals determination by inductively coupled plasma-optical emission spectrometry (ICP-OES) in their ashes and the health risk assessment for the operators. Microchemical Journal, 127, 178-183
Author Response
Comment 1
Page 9 line 296
Based on the previous studies, microwave-assisted (MAE) is an extraction method adopted to obtain organic materials and pesticide remnants from natural products [129].
Please, add in the sentence microwave-assisted (MAE) is an extraction method adopted to obtain organic materials, inorganic and pesticide remnants from natural products [129].
Answer: The sentence is modified as suggested in lines 295-297. Page 9
Comment 2
In this context please add reference
Wood pellets for home heating can be considered environmentally friendly fuels? Heavy metals determination by inductively coupled plasma-optical emission spectrometry (ICP-OES) in their ashes and the health risk assessment for the operators. Microchemical Journal, 127, 178-183
Answer: The reference added and the number of this reference is 130

Reviewer 2 Report
The file is attached below.

Author Response
Comment 1
This work is comprehensive in the sense that the authors summarize the most important literature information about the synthesis of MNPs using Eucalyptus species. Moreover, the factors that influence the production process, characterization techniques and potential recent applications are also mentioned. I think the reference list is quite long. However, because it is a summary of the information, it is probably all right. In chapter 5, some results could be shown, not only cited. It is very difficult for readers to follow all the references. Answer: Thanks for your valuable review. In terms of some results in chapter 5, the clear results have been added in lines 450 - 451, lines 455 – 459 and 453 - 454. Comment 2 Line 20 – Replace scientists´ by scientist´s as well as in line 199. Answer: scientist´s has been added in lines 20 and 199.Page 1 and 6. Comment 3 Line 27 and line 36 - characterisation versus characterization... British or American English? Please, revise English in the whole manuscript. Answer: The whole paper has been revised in according to American English. Comment 4 Line 19 and line 20 – Replace „as presented in (Figure 3)” by “as presented in Figure 3”. Answer: Correction has been done. Comment 5 Line 141 – Eucalyptus genus or genus Eucalyptus. Answer: genus Eucalyptus has been replaced by Eucalyptus genus in line 142. Page 5 Comment 6 Line 210 –Spacing between “]” and “while” must be added. Answer: Correction has been done Page 6 Comment 7 Line 281 – Subcritical water is mentioned above in line 279. It would be better to put the abbreviation SW in line 279. Answer: The abbreviation SW has been added in line 298. Page 9. Comment 8 Line 291 – 300°C instead of 300 °C Answer: correction has been done. Page 9 Comment 9 Line 533 – UV-vis instead of Uv-vis. Answer: Correction has been done. Page 14 Comment 10 Sometimes, it is written Eucalyptus and sometimes Eucalyptus, why? Please, review this in the whole manuscript. Answer: All Eucalyptus terms have been revised in Italic .
